# Degradable Pure Magnesium Used as a Barrier Film for Oral Bone Regeneration

**DOI:** 10.3390/jfb13040298

**Published:** 2022-12-15

**Authors:** Xianfeng Shan, Yu Xu, Sharafadeen Kunle Kolawole, Ling Wen, Zhenwei Qi, Weiwei Xu, Junxiu Chen

**Affiliations:** 1Department of Stomatology, The Affiliated Xinhua Hospital of Dalian University, Dalian 116000, China; 2Key Laboratory of Materials Surface Science and Technology of Jiangsu Province, Changzhou University, Changzhou 213164, China; 3Mechanical Engineering Department, School of Engineering and Technology, Federal Polytechnic, Offa P.M.B 420, Nigeria

**Keywords:** bone regeneration, cytotoxicity, degradation, MAO, pure Mg membrane

## Abstract

The barrier membrane plays an extremely critical role in guided bone regeneration (GBR), which determines the success or failure of GBR technology. In order to obtain barrier membranes with high mechanical strength and degradability, some researchers have focused on degradable magnesium alloys. However, the degradation rate of pure Mg-based materials in body fluids is rather fast, thus posing an urgent problem to be solved in oral clinics. In this study, a novel micro-arc oxidation (MAO) surface-treated pure Mg membrane was prepared. Electrochemical tests, immersion experiments and in vivo experiments were carried out to investigate its potential use as a barrier membrane. The experimental results showed that the corrosion resistance of a pure Mg membrane treated by MAO is better than that of the uncoated pure Mg. The results of cell experiments showed no obvious cytotoxicity, which suggests the enhanced differentiation of osteoblasts. At the same time, the MAO-Mg membrane showed better biological activity than the pure Ti membrane in the early stage of implantation, exhibiting relatively good bone regeneration ability. Consequently, the MAO membrane has been proven to possess good application prospects for guided bone regeneration.

## 1. Introduction

In recent years, with the increasing maturity of oral implant technology, the success rate of oral implants has been significantly improved. Compared with traditional fixed bridge restoration, the main advantage of dental implants is that they do not pose mechanical damage to the natural teeth on both sides of the missing tooth and also possess the advantages of fixed bridge restoration, such as aesthetics, comfort and high chewing efficiency. However, the lack of alveolar bone caused by the premature loss of teeth is a major obstacle to the application of dental implant technology. The guided bone regeneration (GBR) technique is used to facilitate bone regeneration, which uses a biocompatible membrane acting as a physical barrier to prevent the adjacent connective tissue from invading the bone defect in order to achieve the purpose of full osteogenesis [1].

The barrier membrane plays an important role in GBR technology, which almost determines its success or failure. At present, there are many kinds of barrier membrane materials, and their performances are different. They can be roughly divided into two kinds: non-absorbable membranes and absorbable membranes [2]. Non-absorbable membranes include the titanium membrane, which is the most widely used in clinics, and its efficacy has been affirmed [3,4,5]. However, non-absorbable membrane materials possess some inherent deficiencies such as plastic difficulties, membrane displacement and the need for secondary or revised surgery. In view of these deficiencies of non-absorbable membranes, absorbable membranes are favored by an increasing number of clinicians and patients [6]. At present, absorbable membranes are mainly classified into polymers and animal collagens according to their composition. They can be directly absorbed by the human body without a second operation. However, the defects of absorbable membrane-shaped materials are also evident, such as an unstable degradation rate and their inability to provide appropriate volume stability for the bone graft area due to their low mechanical strength and stiffness. Hence, in developing GBR membrane materials that possess the characteristic high mechanical strength and optimum biodegradability, some researchers have focused on biodegradable magnesium alloys [7,8,9,10]. For instance, Rider et al. [11] implanted the pure Mg membrane into small pigs of the Yucatan Peninsula, and they found that the pure Mg membrane degraded faster in vivo. Meanwhile excessive air pockets may cause inflammation and negatively affect the formation of new bones. Observing through micro-CT, it was found that the pure Mg membrane could basically maintain the original shape and position in the first 4 weeks, but by week 8, the pure Mg membrane had severely corroded, preventing its segmentation. Consequently, researchers have adopted some physical or chemical surface modification methods to reduce the corrosion rate of Mg alloys, such as the plasma spraying method, electrochemical method (the deposition method, plasma electrolytic oxidation method, etc.) [12,13], cold spraying method, MAO, etc. [14,15]. Kacarevic et al. [16] prepared MgF_2_ coating on the surface of WZM211 Mg alloy by the chemical deposition method. After implantation in Yucatan minipigs, it was found that this Mg alloy screw met the standard requirements for guided bone regeneration, and the corrosion rate of the Mg alloys in the body was greatly reduced. In fact, the Mg screws were slowly and completely absorbed after 52 weeks. However, the study found that the coating prepared on the polished Mg alloy substrate by the electrochemical deposition method could easily peel off and be prone to cracking and other phenomena, as the bonding force is poor [17]. Meanwhile studies have shown that MAO coating has strong adhesion and ability regarding the substrate, can produce a dense oxide protective layer, can resist substrate corrosion and possesses good biocompatibility. [18,19,20,21]. Liu et al. [22] successfully prepared calcium metaphosphate (CMP) coatings on the surface of an AZ31B Mg alloy by an MAO-assisted sol-gel method. The results showed that the coated sample could effectively induce the osteogenic differentiation of hBMSC cells. This also provides a basis for our follow-up research on GBR. Therefore, this paper uses the method of MAO to prepare a coating on the surface of Mg so as to further study its potential osteoconductive regenerative ability.

## 2. Materials and Methods

### 2.1. Materials Preparation

#### 2.1.1. Alloy Preparation

The pure Mg ingot (>99.99 wt %) was purchased from Fengfeng Longhai Magnesium Processing Co., Ltd., Handan City, Hebei Province, China. The pure Mg membrane with a length of 1200 mm, a width of 70 mm and a thickness of 0.7 mm was made by rolling. The specific processing route is as follows: the initial as-cast plate specification is 400 mm × 150 mm × 5 mm. First, the indoor temperature and humidity were kept at 20 °C and 20%, respectively, and the as-cast plates were homogenized and heat-treated at 300 °C for 2 h and then uniformly rolled into 2 mm-thick membrane materials, which were then heat-treated at 200 °C for 2 h. The specific process parameters were a holding time of 5 min, a rolling force of 3 metric tons, a rolling speed of 2 m/min, an oil temperature of 250 °C, a roll diameter of 80 mm upper and lower rolls and a rolling pass of one time; no annealing was performed after rolling. Finally, a number of square samples with a side length of 8 mm (for the microstructure observation and in vivo experiment) and circular samples with a diameter of 6 mm (for in vitro experiments) were prepared by wire cutting. The samples were polished by 2000 grit^#^ SiC sandpaper step by step to a 0.6 mm thickness. The samples were then placed in anhydrous ethanol, cleaned by an ultrasonic shock machine for 10 min and then taken out and dried for later use. The pure titanium membrane for the in vivo experiment and the micro pure titanium nail for the fixed membrane were provided by Xi’an Zhongbang Titanium Biomaterials Co., LTD, Xi’an, China. The nail cap diameter, nail diameter and length were 2.50 mm, 0.45 mm and 1.23 mm, respectively.

#### 2.1.2. Coating Fabrication

MAO coatings were prepared by the WHD-20 MAO machine (Harbin University of Technology, Harbin, China). The electrolyte contained 1.2 g/L Ca (OH)_2_, 4 g/L Na (PO_3_)_6_ and 8 g/L KF, with a working frequency of 1000 Hz, a duty cycle of 40% and a working time of oxidation of 5 min. After the preparation of the coating samples, they were first rinsed with water and then anhydrous ethanol and later dried in air.

### 2.2. Microstructural Characterization

The surface morphology and analytical element composition of the samples were observed by SEM SU3500 (Hitachi, Hitachi Hi Tech, Japan) and energy dispersive X-ray spectroscopy (EDS), respectively.

### 2.3. Electrochemical Test

Gamry Instruments (Reference 600) were used for electrochemical testing in Hank’s solution. A typical three-electrode system was used for the electrochemical test. Before the test, an open circuit potential (OCP) measurement was carried out for 30 min to obtain a relatively stable electrochemical system and improve the accuracy of the subsequent measurement. Then, the frequency was adjusted to a range of 100000 Hz–0.01 Hz for the electrochemical impedance spectroscopy (EIS) test. Finally, a scanning rate of 0.5 mV/s was used for the polarization test of the action potential.

### 2.4. Immersion Test

The samples were immersed in Hank’s solution in a centrifuge tube for 14 days. The immersion ratio was 1.25 cm^2^/L. The solution was replaced every day, and the pH value of the solution during the entire immersion period was recorded. After the immersion test, a chromic acid (200 g/L chromic acid and 10 g/L AgNO_3_) solution was used to clean the corrosion products on the sample surface, and the degradation rate was calculated according to the weight loss. The external corrosion rate was calculated according to ASTMG31-12a using Equation (1):
P_*i*_ = (K × ΔM)/(A × T × ρ),
(1)

where P_*i*_ (mm/y) represents the corrosion rate, K represents a constant (8.76 × 10^4^), ΔM represents the mass loss in g, T represents the time of exposure (h), A represents the sample’s area (cm^2^) and ρ represents the sample’s density (ρ = 1.74 g/cm^3^). After soaking the sample for 14 days, the microstructure of the soaked sample was observed by SEM-LSM, and the corrosion mechanism was analyzed.

### 2.5. Cell Tests

#### 2.5.1. Cytotoxicity

Cytotoxicity was evaluated according to the ISO 10993-12 standard. First, 200 μL cell suspension (1 × 10^4^ cells/mL) were inoculated into a 96-well cell culture plate. The suspension was cultured in a constant temperature incubator at 37 °C and 5% CO_2_ for 24 h. The extracts (100 μL) of MAO-Mg, pure Mg and α-MEM (blank control) were added into the three groups of culture plates, respectively. Three time points (24 h, 48 h and 72 h) were selected for the CCK-8 test. The CCK-8 solution (20 μL) was added to each well and incubated in a constant temperature incubator for 4 h. The OD value of each culture well was measured by an enzyme labeling instrument at 450 nm. Each group of samples was measured five times at each time point, and the average value was calculated. The cell viability was calculated using Equation (2).
(2)$$\mathrm{Cell}\;\mathrm{viability}=\frac{{\mathrm{OD}}_{\mathrm{sample}}-{\mathrm{OD}}_{\mathrm{blank}}}{{\mathrm{OD}}_{\mathrm{control}}-{\mathrm{OD}}_{\mathrm{blank}}}\times 100\%,$$

#### 2.5.2. ALP Test

MG63 cells were inoculated into 24-well culture plates with a density of 1 × 10^4^ cells/mL. The cells were cultured using the extract or α-MEM instead of the medium for 24 h. The differentiation behavior of the MG63 cells was evaluated by measuring the ALP activity after 5 and 7 days. The ALP activity was calculated.

### 2.6. In Vivo Test

Twelve male New Zealand rabbits that were 6 months old and 2.5–3.0 kg were used in this experiment. The surgery and treatment of the rabbits were performed strictly according to the regulations and laws of the Standing Committee on ethics in China. Four round bone defect animal models with a diameter of 6 mm were made for each rabbit. Three groups, namely, the MAO-Mg group, Ti group and blank control group, were set in the experiment. The experiment took 2 weeks and 8 weeks as the research time points, and each time point included six experimental rabbits. A total of 48 animal models of round full-thickness bone defects were prepared, and 24 round bone defects were found at each research time point. The experiment was divided into three groups: the MAO pure magnesium film group (Group A), the blank control group (Group B) and the pure titanium film group (Group C). Table 1 shows the grouping of experimental animals, in which AABC, ABBC and ABCC, respectively, represent the grouping after the modeling of the cranial parietal bone of an experimental animal. In order to prevent postoperative infection, intramuscular antibiotics were injected into their legs continuously for 3 days after the animal experiments. Six experimental animals were killed by air embolism through an ear vein injection at the 2nd and 8th weeks after the animal experiments, and a calcein fluorescent labeling solution was injected subcutaneously on the 4th and 3rd days before death. Micro-CT scanning was performed on the specimens obtained after the death of the rabbits, and the BV/TV of each bone defect area was measured. The hard tissue sections made from the animal parietal bone specimens were observed under a laser confocal microscope. Finally, the hard tissue sections were stained with V.G., and the formation of the new bone in each bone defect area was observed under the microscope.

Research manuscripts reporting large datasets that are deposited in a publicly available database should specify where the data have been deposited and provide the relevant accession numbers. If the accession numbers have not yet been obtained at the time of submission, please state that they will be provided during the review. They must be provided prior to publication.

Interventionary studies involving animals or humans, and other studies that require ethical approval, must list the authority that provided the approval and the corresponding ethical approval code.

## 3. Results

### 3.1. Microstructural Characterization

Figure 1 shows the microstructures of the MAO coating. From the cross-section images observation, it can be seen that the thickness of the coating was about 5–10 μm. Moreover, the coating was uniform. For the MAO coating, many micro-pores with a size of 2–5 μm were observed, and the distribution of the micro-pores was uniform. Table 2 shows the EDS analysis of the area marked by A in Figure 1b. The main elements in the MAO coating were Mg, O, F, P and Ca. According to a previous study, the main components of the coating were MgO, HA and MgF [23].

### 3.2. Electrochemical Test

Figure 2 shows the potentiodynamic polarization curves of the pure Mg and MAO-Mg. From the curves, it can be found that the self-corrosion potential of MAO-Mg was much higher than that of pure Mg, indicating that the tendency to be corroded was much lower than that od pure Mg. The more left the curve was, the smaller the corrosion current density was; therefore, it can also be found that the corrosion current density of MAO-Mg was much smaller than that of pure Mg. Table 3 illustrates the Tafel fitting results of the potentiodynamic polarization curves. The corrosion current densities of the pure Mg and MAO-Mg were 5.630 μA/cm^2^ and 0.657 μA/cm^2^, respectively. The corrosion rates of pure Mg and MAO-Mg were 0.324 mm/y and 0.038 mm/y, respectively. The corrosion rate of pure Mg was improved by more than 8 times after the MAO treatment.

Figure 3 shows the EIS curves of the samples. Figure 3a presents the relationship between the real part and the imaginary part of the impedance. For the pure Mg, there is one capacitance loop at the high and middle frequencies and one inductive loop at the low frequency. For the MAO-Mg, there is mainly one capacitance loop. According to previous reports, the diameter of the capacitance loop reflects the corrosion resistance of the samples. The larger the diameter of the capacitance loop, the lower the corrosion rate of the samples. Figure 3b presents the Bode curves of the pure Mg and MAO-Mg. The larger the value of the Bode curve, the better its corrosion resistance. The impedance modulus of the MAO-Mg sample is much higher than that of the pure magnesium sample. In Figure 3c, the phase angle at an intermediate frequency becomes higher and wider, indicating that MAO-Mg samples have a better corrosion resistance. Figure 3d,e show the equivalent circuit of the MAO magnesium sample and the pure magnesium sample, respectively. R_s_ is the solution resistance, R_1_ indicates the resistance of the coating, CPE_1_ and CPE_2_ indicate constant phase elements (CPEs) and R_3_ presents the pitting with inductance L. The fitting results of the equivalent circuit are shown in Table 4. The R_1_ value of the MAO-Mg sample is much higher than that of the pure magnesium sample, which indicates that the MAO coating has effectively protected the matrix and that the MAO-Mg sample has good corrosion resistance.

### 3.3. Immersion Test

The pH change and degradation rate results of MAO-Mg membrane and pure Mg membrane immersed in Hank’s solution for 14 days are shown in Figure 4. It can be clearly observed from Figure 4a that the pH values of both materials showed an upward trend on the first day of immersion, but the pH values of MAO-Mg group were significantly lower than those of pure Mg group. The pH value of the pure Mg group had exceeded 11.5 on the second day, and decreased after 8 days, and decreased to 10.5 on the 12th day. The pH value of MAO-Mg group was 9.5 on the second day of immersion, and kept a steady downward trend every day. It can be seen from the degradation rate diagram (Figure 4b) that the degradation rate of MAO-Mg sample is about 0.16 mm/y, and that of Mg sample is about 0.50 mm/y.

### 3.4. Immersion Experiment

After 14 days of immersion, the pure Mg samples and MAO-Mg samples were taken out and observed after pickling, and their macroscopic corrosion morphologies are shown in Figure 5a,b, respectively. The MAO-Mg sample basically maintains the original morphology, and its coating is still clearly visible. The pure Mg sample was severely corroded, and a small part of its edge had been corroded and peeled off; thus, a large number of pits can be seen on the surface. The morphologies of the corroded samples were further observed by a scanning electron microscope. The corrosion results of the pure Mg sample and MAO-Mg sample are shown in Figure 5c,d, respectively. A layer of corrosion products was formed on the surface of the pure Mg samples, indicating that the samples had been severely corroded. A complete coating can still be observed on the surface of the MAO-Mg sample, showing that there are no substantial corrosion products. This further indicates that the corrosion resistance of the MAO-Mg sample is better.

### 3.5. In Vitro Experiments

#### 3.5.1. Cell Proliferation

Figure 6a shows the OD values of the three groups of MG63 cells detected by CCK-8 at different time points. With the extension of the culture time, the OD values of the three groups were increased. After 24 h, there was no statistically significant difference between the three groups. However, at the 48 h and 72 h time points, the OD value of the MAO-Mg film group was higher than that of the blank group and the untreated Mg group, and the difference is statistically significant. There was no significant difference between the blank group and the Mg group, and MAO-Mg did not show significant cytotoxicity. With the prolongation of the culture time, the proliferation of osteoblasts was better promoted than that of untreated samples.

#### 3.5.2. ALP Activity Measurement

Figure 6b shows the ALP viability of the three groups of MG63 cells on days 5 and 7. On the 5th and 7th day, the activity of ALP in the blank group was the lowest. However, the ALP activity of the MAO-Mg film group was higher than that of the blank group and the untreated Mg group. This suggests that MAO coating can enhance the differentiation of osteoblasts. This result is consistent with the detection result of CCK-8.

### 3.6. In Vivo Experiments

During the preparation of the bone defect, the isolated bone block taken from the defect area is retained to prepare autologous bone powder to fill the defect area. After the defect area was filled with autologous bone powder evenly, the membrane materials were respectively covered in the defect area filled with autologous bone powder. A total of 48 bone defect models were prepared in experimental rabbits and randomly divided into groups. According to the defect area grouping, the covering materials are as follows: the MAO pure magnesium film group, the blank control group and the pure titanium film group. Fix the four sides of the film with micro-nails. Keep a distance of at least 2 mm between the edges of adjacent films. Select diagonal lines for the covering positions of the same film materials. Close the surgical wound and suture it tightly. Figure 7 shows the whole surgical procedure.

The experimental rabbits did not take food on the operation day after the operation, and their mental state was poor. Thus, the amount of activity was significantly lower than that before surgery. However, the experimental rabbits ingested a small amount of food on the first postoperative day. Meanwhile, after three days of observation, the experimental animals have basically returned to a normal diet and activity. There were no infections or deaths in the animals. The wound in the surgical area of the cranial parietal bone of the experimental rabbits healed well. There was no obvious fever or swelling of the soft tissue around the incision, and no bloody exudation was seen. The wound healed at an early stage after a week, and the sutures were found to have fallen off. The skull parietal bone specimens obtained after execution showed that the micro-nails were well fixed without falling off. The area of membranous material did not change significantly at both the 2nd and 8th weeks. However, the thickness of the film gradually decreased with increasing time.

Figure 8 shows the imaging results and changes of BV/TV in the defect area after surgery in each group at each time period. From Figure 8a,c, it can be seen that the 2-week MAO-Mg membrane group and the Ti membrane group had the traces of the modeling during the operation, and some immature bone ossicles could be seen under the membrane beam structure. Figure 8b shows that the hole preparation marks in the modeling area of the blank control group are very obvious, and no mature bone trabecular structure appears. Figure 8d,f also show that there is no significant difference between the images of the MAO-Mg membrane group and the Ti membrane group at 2 weeks, as most of the traces of modeling during the operation disappeared. The number of mature bone trabeculae under the membrane of both groups was significantly increased compared with that at 2 weeks. Meanwhile, at 8 weeks, both trabecular bones were denser and very similar. Figure 8e shows that the blank control group also had mature bone trabecular tissue, and the bone trabecular tissue was denser than that at 2 weeks. However, the density of the trabecular bone in the blank control group was still lower than that in the MAO-Mg membrane group and Ti membrane group.

The image selection operation area after micro-CT reconstruction was analyzed. One-Way analysis of variance (ANOVA) was performed on the BV/TV around the defect area covered by the same material at different time points, and the results are shown in Figure 8g. The pairwise comparison of each group at 2 weeks and 8 weeks showed that the bone mass per unit volume at 8 weeks was greater than that at 2 weeks, and the difference was statistically significant (*p* < 0.05). One-way analysis of variance was performed on the BV/TV of the defect areas of different covering membrane materials at the same time point, and the results are shown in Figure 8h. At 2 weeks after operation, the BV/TV measurement results in the defect area showed that the MAO-Mg membrane group and pure titanium membrane group were significantly larger than the blank control group, and the difference was statistically significant (*p* < 0.05). Furthermore, the MAO-Mg membrane group was larger than the pure titanium film group, and the difference was also statistically significant (*p* < 0.05). At the 8th week after operation, the MAO-Mg membrane and Ti membrane groups were significantly larger than the blank control group, and the difference was again statistically significant (*p* < 0.05). Meanwhile, there was no significant difference between the MAO-Mg membrane group and the pure Ti film group (*p* > 0.05).

As shown in Figure 9, the hard tissue sections were placed under the laser confocal microscope to observe the irregular band-shaped and circular calcein-labeled areas, and they appeared to be fluorescent green under the microscope. It can be seen that, 2 weeks after surgery, all three groups showed weaker fluorescent staining. There was no significant difference in the fluorescent staining between the MAO-Mg membrane group and the Ti membrane group. The fluorescent bands were brighter and continuous compared to the blank control group. Compared with the blank control group, the MAO-Mg membrane group and the Ti membrane group showed stronger fluorescent staining of the new bone tissue under the membrane. At the 8th week after operation, the fluorescent staining of each group was significantly enhanced compared with that of the 2-week group. The fluorescent staining othe f MAO-Mg membrane group and the Ti membrane group was brighter, more intense and more continuous. However, the fluorescence staining of the blank control group was relatively weak and distributed.

Hard tissue sections of each group were stained with Van Gieson’s picric acid–acid fuchsin and observed under a light microscope. There is new bone formation in the defect area (shown in red color), as shown in Figure 10. It can be seen that, 2 weeks after the operation, the blank control group still showed granular bone tissue, which was relatively scattered. Relatively continuous bone tissue appeared in the MAO-Mg membrane group and the Ti membrane group. Eight weeks after the operation, a sheet-like new bone tissue was seen on the edge and in the center of both the MAO-Mg membrane group and the Ti membrane group. Some of the cavities near the edge of the material had overgrown with new bone tissue, probably because there was more blood and interstitial fluid exchange at the edge of the material. There was significantly less new bone tissue in the blank control group than there was in the MAO-Mg membrane group and the Ti membrane group. However, compared with 2 weeks, all three groups had more continuous and abundant bone tissue filling.

## 4. Discussion

Degradable metal Mg film has good biological safety and excellent mechanical properties, which makes its gradual application in GBR a continuous one [24,25,26,27]. The collagen barrier membrane is also a common absorbable barrier membrane which possesses high biological activity. During degradation, the problem of membrane collapse occurs frequently. The excellent mechanical properties of degradable magnesium film can solve this problem [28,29,30]. Previous studies have shown that pure magnesium barrier films meet the necessary requirements for mechanical and biological safety in the treatment of GBR [11]. However, the degradation rate of magnesium is not controlled, and, as such, osteolysis easily occurs, which invariably does not coincide with the time of osteogenesis and thus hinders its wide application [14,31].

In order to solve this problem, we treated the surface of pure magnesium to reduce its degradation rate. After the application of the MAO on the pure magnesium surface, evenly distributed micropores appeared on them. According to the EDS results, the main components of the coating were MgO, HA and MgF_2_. Previous studies have shown that the preparation of MgF_2_ on the surface of the magnesium matrix can effectively reduce its corrosion rate, and the corrosion rate is about 20 times higher than that of untreated samples [24,32]. The electrochemical data in this study show that the corrosion rate of MAO-Mg is decreased by about 90% compared with that of pure Mg. Lin et al. [33] found that, during the degradation process, the slow degradation rate can release the appropriate magnesium ion concentration (of about 13.2 mM). This can not only improve the activity and proliferation of cells but also help wound healing. The lower degradation speed can make the human tissue gradually process or discharge the biodegradation products and reduce the occurrence of adverse reactions [34].

To better understand the in vitro degradation behavior of the two samples, immersion experiments were performed. The experimental results showed that the pH values of the pure-Mg samples were higher than those of MAO-Mg. Meanwhile, since the surface of the pure-Mg sample was not protected by coating, a cathodic reaction occurred, releasing OH- ions and resulting in a higher pH value [35]. During the 14-day immersion experiment, the pH of the pure-Mg samples did not increase continuously, which may be due to the easy formation of stable hydroxides at high pH values. The sparse hydroxide layer can block the contact between part of the electrolyte and the substrate, thus improving its corrosion resistance [36,37]. The corrosion resistance of MAO-Mg is better than that of the pure-Mg samples because the dense coating can delay gas evolution during its degradation [38], thereby reducing the generation of air pockets and aiding in the integration of new bones after implantation in the human body [25].

In vitro experiments have demonstrated that the MAO coating obviously improved the corrosion resistance of Mg samples. However, our ultimate research objective was to develop a magnesium membrane with a suitable degradation rate and good biosafety. Therefore, the samples were tested for cytotoxicity and ALP in vitro. A previous study showed that MAO coatings prepared with the same electrolyte do not only promote the growth and reproduction of cells but also displayed the highest cell viability [23]. The results of these experiments also showed that, with the increase in culture time, the MAO coating could easily promote the proliferation of osteocytes. It can be seen from Figure 5a that the cell viability index OD value of MAO-Mg at 72 h is the highest. The higher the OD value, the greater the viability of the cells and the stronger their proliferation ability. The ALP activity test is one of the most commonly used methods to detect early osteogenic differentiation. The higher the ALP activity value, the better the osteogenicity. The ALP activities of the MAO-Mg group were always higher than those of the other two groups, and the differences were significant. Therefore, the osteogenic effect of the MAO-Mg group is better. Some studies have also shown that the corrosion products produced by magnesium alloys during the degradation process have antibacterial effects on three common bacterial strains (Escherichia coli, Pseudomonas aeruginosa and Staphylococcus aureus) and can equally improve the activity of bone cells [39].

Magnesium alloy has good biosafety, exhibits no chronic inflammation or other negative phenomena after implantation and also possesses good bone regeneration ability [40,41].

Hence, to further analyze their osteogenic ability in vivo, two samples were implanted in New Zealand rabbits in this study. The BV/TV ratio and BS/BV ratio were used to evaluate the osteogenic ability. Higher BV/TV ratios indicate a higher bone volume, and lower BS/BV ratios indicate a higher bone structure density. Taking Ti and the blank control as the reference, the BV/TV ratio of the MAO-Mg group was the same as that of the Ti group and higher than that of the control group. This indicates that the volume of new bone formation in the MAO-Mg group was higher than that in the blank control group and was similar to that in the Ti group, as shown in Figure 6. However, the BMD of the MAO-Mg group was the lowest, which may be caused by osteolytic reaction at the implantation site or incomplete mineralized callus [42]. The imaging scan results of the MAO-Mg and Ti at 8 weeks showed a mature and compact trabecular bone structure. Previous studies have shown that Mg alloys have a positive effect on the increase in new bones [43,44,45], and this is also consistent with our findings.

This experiment evaluated the in vitro biosafety, degradation process and in vivo guided bone regeneration of magnesium films with MAO coating. However, its in vitro mechanical properties and in vivo degradation process have not been evaluated due to the limited number of animal samples. In the follow-up experiments, in vitro mechanical experiments and in vivo biodegradation tests will be conducted.

In the overall analysis, MAO-Mg has a suitable degradation rate and biosafety, as well as an excellent ability to guide bone regeneration, which meets the requirements for application in GBR. In addition, MAO-Mg is more suitable for clinical applications. Thus, in order to make them more effective in improving the bone regeneration ability, further research is underway.

## 5. Conclusions

In this paper, the in vitro degradation of pure magnesium membranes coated with MAO coating was investigated, and the cytotoxicity was analyzed. At the same time, combined with in vivo experiments, the effect of MAO-Mg on the new bone formation was studied in depth, and the following conclusions were drawn:

(1) The corrosion resistance of MAO-Mg was better than that of uncoated pure magnesium. After 14 days of immersion, the degradation rate of the MAO-Mg sample was about 0.16 mm/y, which was about one-third of that of the pure magnesium sample.

(2) MAO-Mg did not show obvious cytotoxicity, and the ALP activity was higher than that of the blank group and the untreated Mg group. MAO coating could enhance the differentiation of osteoblasts.

(3) The MAO-Mg membrane showed better biological activity than the pure titanium membrane in the early stage of implantation (within 2 weeks after the operation) and showed relatively better bone regeneration ability. At 8 weeks after the operation, the bone regeneration abilities of the pure magnesium membrane and the pure titanium membrane were similar due to the exposure of the magnesium membrane. Therefore, the pure magnesium film whose surface is treated by MAO has good application prospects for guided bone regeneration.

## Figures and Tables

**Figure 1 jfb-13-00298-f001:**
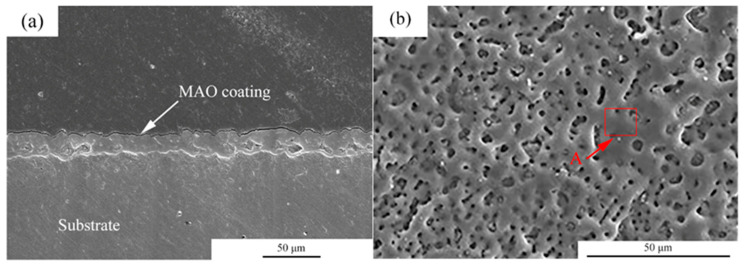
Microstructures of the MAO coatings, (**a**) cross-section morphologies, (**b**) surface morphologies.

**Figure 2 jfb-13-00298-f002:**
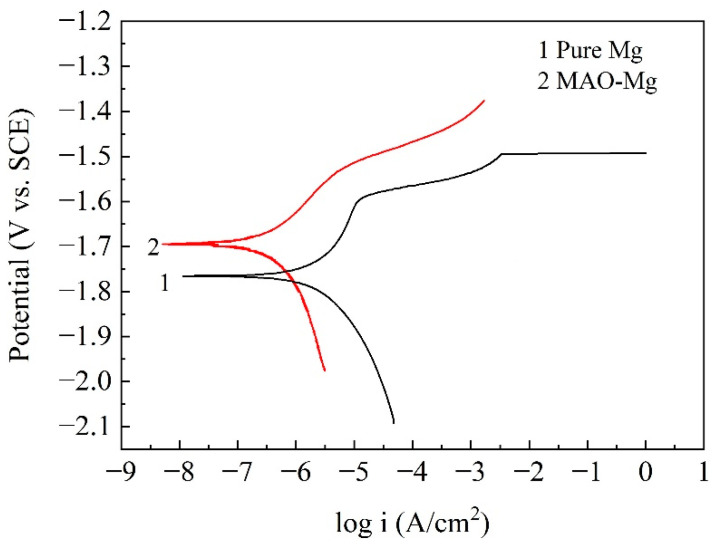
Potentiodynamic polarization curves of the samples in Hank’s solution.

**Figure 3 jfb-13-00298-f003:**
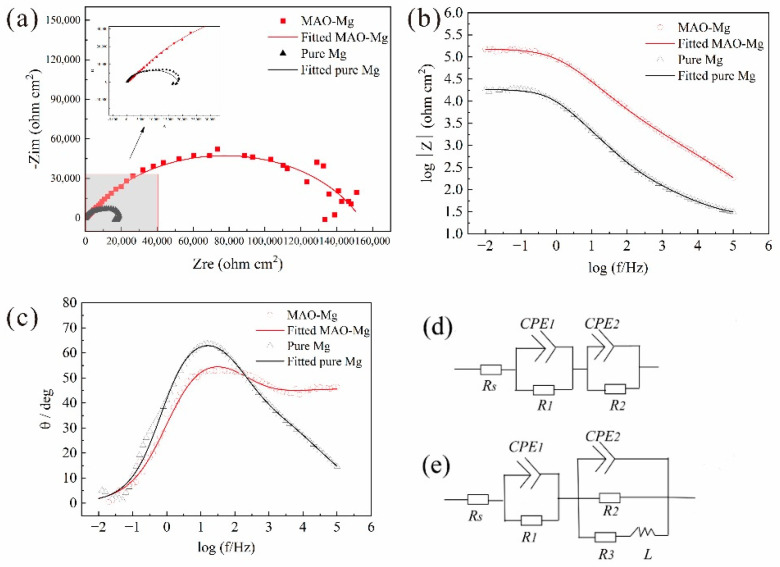
EIS curves, (**a**) Nyquist curves, (**b**) Bode curves, (**c**) Bode phase angle, (**d**) Equivalent circuit of the MAO-Mg sample and (**e**) Equivalent circuit of the pure Mg sample.

**Figure 4 jfb-13-00298-f004:**
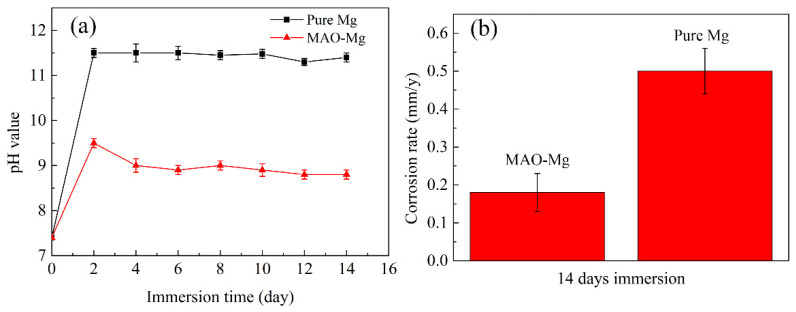
Immersion tests of (**a**) the pH change in Hank’s solution immersed with the alloy for a different time and (**b**) the corrosion rates of the alloy after 14 days of immersion in Hank’s solution.

**Figure 5 jfb-13-00298-f005:**
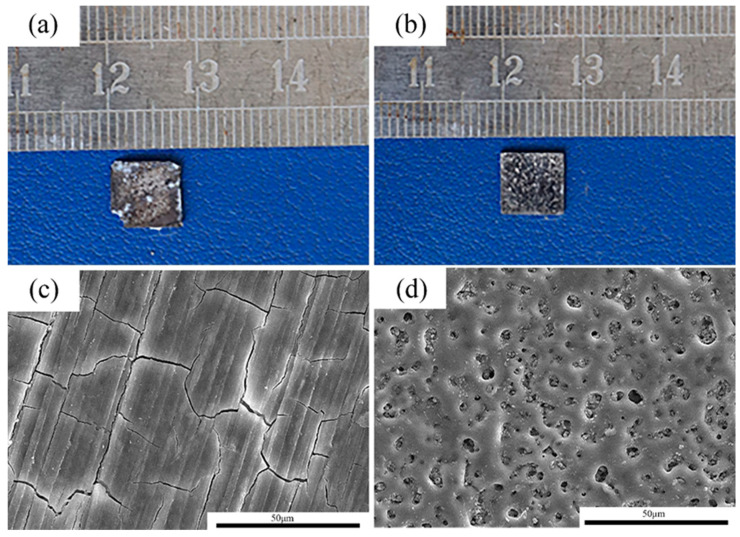
The corrosion morphologies of the alloy after 14 days of immersion in Hank’s solution. (**a**,**c**): pure Mg; (**b**,**d**): MAO-Mg.

**Figure 6 jfb-13-00298-f006:**
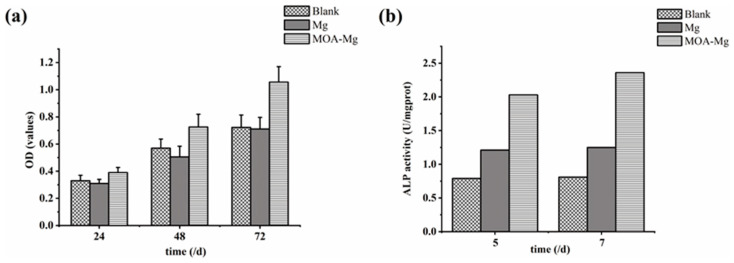
(**a**) CCK-8 results of the growth of the three groups of MG63 cells; (**b**) ALP activity of the three groups of MG63 cells.

**Figure 7 jfb-13-00298-f007:**
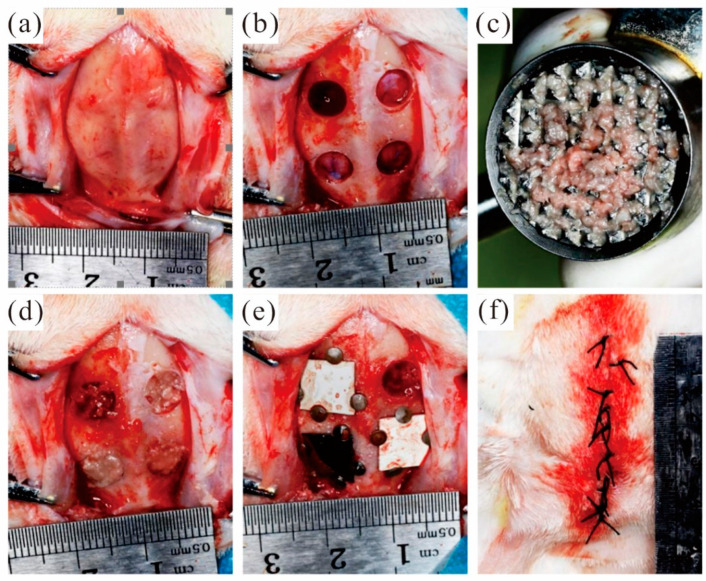
Animal experimental operation process: (**a**) open the soft tissue and periosteum of the skull top; (**b**) The round bone defect area was prepared; (**c**) Grind the autologous bone block into autologous bone powder; (**d**) Autogenous bone powder was used to fill the defect area; (**e**) Cover the film and fixation; (**f**) Suture.

**Figure 8 jfb-13-00298-f008:**
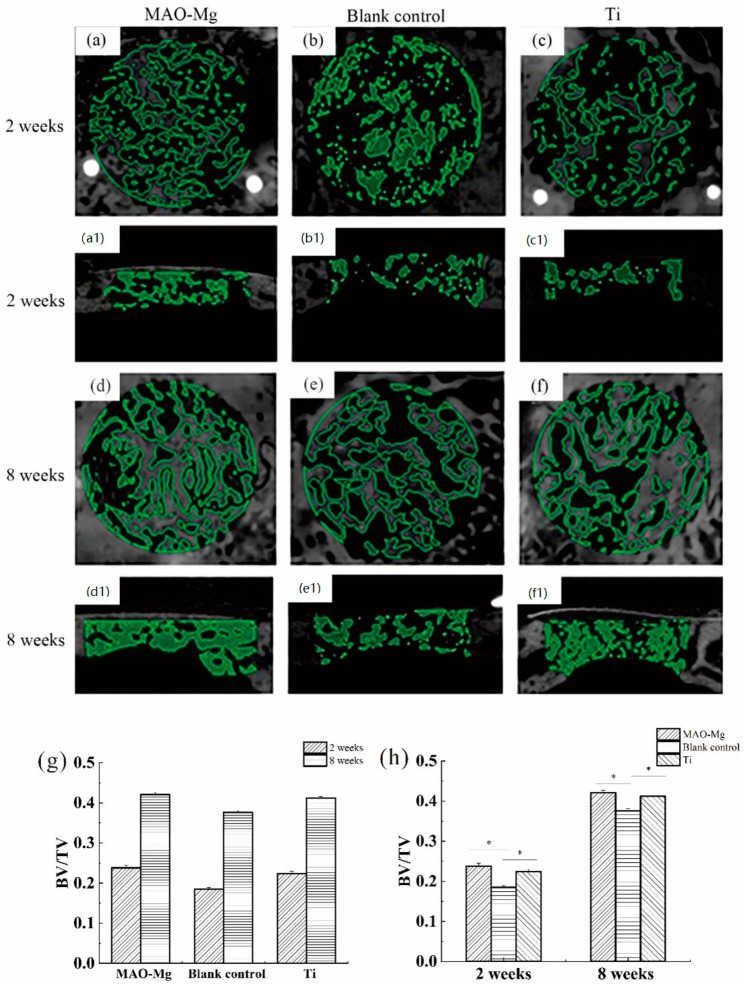
Images of micro-CT after 3D reconstruction and changes in BV/TV in the defect area after surgery, (**a**,**a1**) 2-week MAO pure magnesium membrane group and Osteogenic height of bone defect, (**b**,**b1**) 2-week blank control group and Osteogenic height of bone defect, (**c**,**c1**) 2-week pure titanium membrane group and Osteogenic height of bone defect, (**d**,**d1**) 8-week MAO pure magnesium membrane group, (**e**,**e1**) 8-week blank control group and Osteogenic height of bone defect, (**f**,**f1**) 8-week pure titanium membrane group and Osteogenic height of bone defect, (**g**) Changes in BV/TV in the defect area at different time points after operation; (**h**) Changes in BV/TV in the defect area after different treatments in the same time period (* *p* < 0.05).

**Figure 9 jfb-13-00298-f009:**
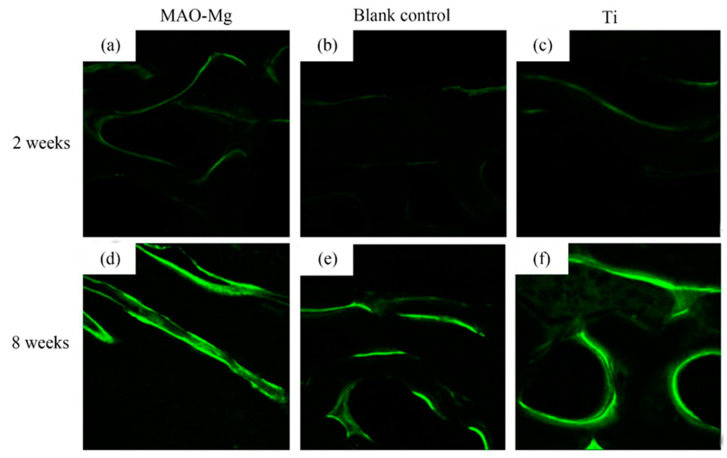
Laser confocal microscope observation (×100), (**a**) 2 weeks MAO pure Mg membrane group, (**b**) 2 weeks blank control group, (**c**) 2 weeks pure Ti membrane group, (**d**) 8 weeks MAO-Mg membrane group, (**e**) 8 weeks blank control group, (**f**) 8 weeks pure Ti membrane group.

**Figure 10 jfb-13-00298-f010:**
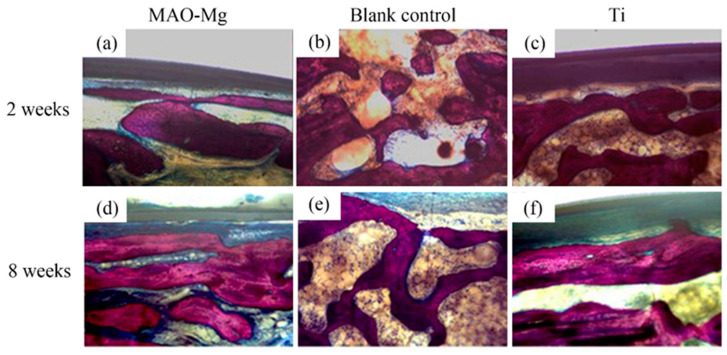
The bone slices with implants under optical microscope V.G. staining (×100).

**Table 1 jfb-13-00298-t001:** The number of experimental animals was counted according to the model of the parietal bone in each time period (piece, n = 12).

Time Slot	Group			Total
	A A B C	A B B C	A B C C	
2 weeks	2	2	2	6
8 weeks	2	2	2	6

**Table 2 jfb-13-00298-t002:** EDS analysis of the areas marked by A in Figure 1.

Position	Composition (at.%)				
	O	F	Mg	P	Ca
A	51.31	7.03	27.78	12.10	1.78

**Table 3 jfb-13-00298-t003:** Tafel fitting results of the potentiodynamic polarization curves.

Samples	*i*_corr_ (μA/cm^2^)	*E* (V)	Corrosion Rate (mm/y)
Pure Mg	5.630	−1.760	0.324
MAO-Mg	0.657	−1.690	0.038

**Table 4 jfb-13-00298-t004:** Fitting results of the alloy from the impedance curves in Hank’s solution.

Materials	*R*_s_ (Ω cm^2^)	*CPE* _1_		*R*_1_ (Ω cm^2^)	*CPE* _2_		*R*_2_ (Ω cm^2^)	*R*_3_ (Ω cm^2^)	*L* (H cm^−2^)
		*Y*_01_ (μΩ^−1^ cm^−2^ s^−1^)	*n* _1_		*Y*_02_ (μΩ^−1^ cm^−2^ s^−1^)	*n* _2_			
Pure Mg	21.99	149.30	0.48	140.6	16.86	0.80	20.35 × 10^3^	198.2 × 10^3^	4.45
MAO-Mg	3.72 × 10^−3^	9.95	0.48	1.90 × 10^3^	1.79	0.71	151 × 10^3^	-	-

## Data Availability

Not applicable.

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
