# Peer review of "Degradable Pure Magnesium Used as a Barrier Film for Oral Bone Regeneration"

_jfb, 2022, doi:10.3390/jfb13040298_

Round 1
Reviewer 1 Report
I salute the authors who conducted many experiments.
However, it seems that this manuscript will have to go through some revisions.
1. Some typos are found in the English sentences of the manuscript, and I feel that the readability is poor. It seems that a correction is needed on this point.
2. The chemical composition of Hank's solution is well known. But why the authors show the composition as table 1?
3. The description of Figure 7 is given in the manuscript, but the sentence describing Figure 7 is not observed in the manuscript.
4. If the authors did a statistical analysis, it would be good to implement a visual notation for significance in Figure 7.
5. A photograph of the Mg membrane sample can only be observed in Figure 5. However, it is necessary for authors to present photos showing the application in the in vivo study. It may be able to cover the 6mm defect, but it seems to be able to move because there is no fixing screw and the size is not large.
6. In an in vivo study, it was said that four 6mm diameter holes were created per rabbit. However, it is estimated that group assignments were made to each hole of three groups. How did the authors do the remaining 1 hole?
7. The y-axis in Figure 7 is BV/TV in percentage. But is the 0.1-0.5 notation accurate?
Author Response
Dear Professor, thank you for kind your suggestion. These opinions are valuable and helpful for revising and proving our papers. We have studied it. All comments are careful and have been carefully corrected. The revised part is marked in the article.

Reviewer 2 Report
Please see enclosed file.

Author Response

(The authors gave the same response as above.)

Round 2
Reviewer 1 Report
I appreciate the efforts of the authors for revision.
Author Response
Dear editor: Dear editor, thank you for kind your suggestion. These comments will improve the quality of the manuscript. We examined our manuscript carefully. It further clarifies the logic of writing and the quality of manuscripts. We have modified the grammar in the article. Please see the latest uploaded manuscript. Kind regards, Yu XuReviewer 2 Report
The authors take in consideration all the suggestions.
Author Response
Dear editor: Dear editor, thank you for kind your suggestion. These comments will improve the quality of the manuscript. We examined our manuscript carefully. It further clarifies the logic of writing and the quality of manuscripts. We have modified the grammar in the article. Please see the latest uploaded manuscript. Kind regards, Yu Xu